# Challenges in Assessing the Behaviour of Nodal Electricity Prices in Insular Electricity Markets: The Case of New Zealand

Daniela Pereira Macedo [1,2], António Cardoso Marques [1,2,*] and Olivier Damette [3,4]

1    Management and Economics Department, University of Beira Interior, 6201-001 Covilhã, Portugal
2    Department of Management and Economics, NECE-UBI—Research Unit in Business Science and Economics, University of Beira Interior, 6201-001 Covilhã, Portugal
3    BETA-CNRS, University of Lorraine, 54035 Nancy, France
4    BETA-CNRS, University of Strasbourg, 67081 Strasbourg, France
*    Correspondence: amarques@ubi.pt or acardosomarques@gmail.com

**Abstract:** In this new era of energy transition, access to reliable and correctly functioning electricity markets is a huge concern for all economies. The restructuring path taken by most electricity markets involves the movement towards green generation structures and the increasing integration of wind and solar photovoltaic energy sources. Furthermore, it involves the electrification of energy systems, which implies a substantial increase in electricity demand levels. It is also important to add that electricity use has been pivotal in achieving efficient productivity levels in many sectors and is thus crucial to boosting economic activity. Nevertheless, this shift in generation structures has raised several challenges in electricity markets, mainly because the electricity produced from wind and solar photovoltaics is intermittent. In turn, adopting green power sources has been claimed to increase electricity price volatility and thus increase pricing risks. Therefore, to ensure that the right market signals are being sent to investors, the behaviour of electricity prices should be carefully assessed. There are three main types of pricing mechanisms commonly used in electricity markets: zonal, uniform and nodal. This study provides a short literature survey on these three pricing mechanisms. Our analysis has revealed that the assessment of the behaviour of nodal electricity price volatility is rarely studied in the literature. This fact has motivated the exploration of this topic and the consideration of the New Zealand electricity market case. The New Zealand electricity market is an energy-only system with no interconnections with other electricity markets. Furthermore, it has plenty of electricity produced from hydropower, which has a high potential to reduce price volatility through its backup role. The nodal pricing mechanism is complex, and data on it are hard to process. This paper elucidates the main challenges in processing electricity big data. Three different procedures to make this data more useable are described in detail. The main conclusions of this paper highlight the need to access easy-to-manage data and identify certain variables that significantly affect nodal prices for data which are unavailable.

**Keywords:** insular electricity systems; nodal pricing; New Zealand; renewable energy sources

## 1. Introduction

The path to the decarbonisation of the electricity sector has been marked by the increasing use of wind and solar photovoltaic (PV) energy sources. The electricity produced from renewable energy sources (RES) is highly dependent on weather conditions, and thus, is frequently intermittent. Furthermore, they are likely to cause a decrease in electricity prices as their marginal costs are nearly zero, also known as the Merit-Order Effect (MOE). Moreover, the large-scale electricity storage capacity is still low. Therefore, electricity markets still require robust and flexible backup generation to satisfy electricity demand during periods of low production by renewable sources. To further ensure stable electricity prices and enhance the security of supply, some electricity systems have also established

cross-border interconnections to harness surpluses in neighbouring systems. In many countries, these strategies can mitigate major disruptions in wholesale electricity prices, such as volatility, seasonality, and price spikes, and bring benefits from economies of scale, the diversification of energy sources, and consumption patterns. In this new era of energy transition which entails the electrification of energy systems and an increase in electricity consumption, the careful assessment of electricity market dynamics is indeed crucial to ensure the correct functioning of economic activity.

In isolated electricity systems, such as those on islands, establishing interconnections with other electricity systems is not always a feasible/easy option to implement. These insular electricity systems are often less reliable, as they are generally: (i) highly dependent on overseas trade (especially expensive imported fuels); (ii) unable to exchange electricity with other electricity markets; and hence (iii) less likely to have the backup capacity needed to handle intermittent RES generation (Erdinc et al. 2015). If insular systems remain highly dependent on imported fossil-fuels, the costs of electricity generation and the storage of primary energy sources tend to be much higher than those of interconnected (mainland) systems. Therefore, the electricity produced from RES could provide substantial benefits for insular electricity systems and could be pivotal in reducing electricity production costs. However, integrating intermittent RES into insular systems raises its own challenges. In particular, efficiently managing schedules and market operations to avoid blackouts requires considerable effort (Simoglou et al. 2016). Thus, in the absence of better alternatives, the consumption of excess RES generation must be deferred, and this requires providing extra storage capacity, such as hydroelectric pumped storage.

One of the most distinctive insular electricity markets is that of New Zealand (NZ). One of the characteristics of the NZ electricity market is its extensive portfolio of RES generation and correspondingly low reliance on fossil fuels. However, its electricity system has no price caps and/or capacity market to help it deal with unexpected shocks in electricity prices. The NZ electricity system is dominated by hydro-power, but its storage capacity is limited. Consequently, during dry periods, electricity price spikes are common. Furthermore, unlike most electricity markets, such as those in Europe, it rarely offers incentives to increase the installed capacity of RES. According to the NZ Electricity Authority, its main strategy is to increase competition by reducing entry barriers and promote competitive markets. Nevertheless, there is evidence that significant market power is being exercised in this electricity system (Poletti 2021), which is not surprising given the amount of investment required (mainly related to hydropower generation capacity) to enter this particular market. In fact, according to the data published on the official website of NZ's Electricity Authority, five major electricity producers hold around 92% of the total operational capacity.

The electricity market in NZ uses a nodal pricing mechanism. In this pricing system, the market equilibrium price is set by considering electricity consumption and production levels, and the associated costs of electricity transmission (i.e., losses and congestion in transmission lines within nodes). This should ensure the overall efficiency of the nodal electricity system and increase the transparency of electricity pricing. However, during periods of transmission congestion, price spreads between nodes are likely to occur. In the case of NZ, this could be a concern since there is an electricity production surplus in the North Island (NI) and a consumption deficit in the South Island (SI).

The efficient integration of wind and solar PV power in the generation mix of any electricity system is still challenging. Their intermittent generation increases electricity price volatility in many electricity markets (Macedo et al. 2021; Rintamäki et al. 2017; Martinez-Anido et al. 2016), and makes electricity prices less predictable. The impact of intermittent RES on the dynamics of electricity prices must be assessed and compared in several electricity markets with different generation mixes, pricing mechanisms, and electricity consumption patterns. This would enable consistencies in electricity markets to be identified, and thus, more efficient measures to be developed for integrating RES into electricity markets.

There is a notable lack of empirical studies on the impact of RES on electricity price volatility in systems with nodal pricing mechanisms. Therefore, the effectiveness of the nodal electricity pricing mechanism in smoothening the well-known volatility of electricity price while integrating RES in electricity systems is still being discussed little in the literature, the case of the NZ electricity market in particular. To the best of our knowledge, the study of Wen et al. (2022) is the most recent analysis of the NZ electricity market. It analysed the MOE, but only studied a brief period from 2011 to 2012, so it is somewhat outdated. Given this, why has the NZ electricity market not been the subject of empirical analysis? The NZ electricity market has interesting features that deserve further discussion.

The main contribution of this paper to the literature is twofold. First, it provides a short critical literature survey related to the three main electricity pricing mechanisms, namely: uniform, nodal, and zonal. It discusses the dynamics of electricity markets under different pricing mechanisms. The main aim is to compare the impact of wholesale electricity prices on the electricity produced from wind and solar PV power under different pricing regimes. This analysis aims to identify and discuss the most efficient strategies in different electricity systems to promote the diversification of the generation mix. In fact, given the rise in electricity demand levels, this commodity is increasingly playing a crucial role in boosting productivity in most sectors, and hence is increasing economic activity. The second main contribution is related to the process of obtaining nodal electricity price big data. This study initially sought to analyse the well-known MOE and assess the behaviour of the volatility of the NZ electricity price. However, the formation of nodal electricity prices is highly complex; thus, filtering and downloading data from nodal electricity pricing systems is computationally challenging and demanding. During the process of building the database from the NZ electricity market, numerous problems, such as gaps in the files and mismatches between them, were found. Several procedures were followed to produce a database with as few bugs as possible to make an econometric analysis feasible and reliable. Therefore, this paper briefly describes the main constraints that thwarted an econometric analysis of the MOE from wind power in the New Zealand electricity market from 2012 onwards. The various procedures and options chosen to achieve a consistent and detailed analysis are described in full. We trust that the practices described in this paper will provide valuable guidance for future research on other insular systems with nodal electricity prices.

Overall, this paper aims to fill a clear gap in the literature by discussing the specific characteristics of various electricity pricing mechanisms, with a particular focus on nodal electricity pricing in the insular system of NZ. Furthermore, considering the challenges we faced in processing the data, it describes in detail the different procedures followed to obtain a database with as few bugs as possible. Although we were unable to follow an established econometric procedure, we were able to identify the urgent need for more usable data on nodal electricity pricing. We also conclude that a benchmarking analysis of the strengths and weaknesses of the three pricing mechanisms most common in electricity markets worldwide is essential.

The remainder of this paper is structured as follows: Section 2 provides a brief review of the three main pricing systems; Section 3 presents the challenges faced when studying nodal electricity prices and considers the specific case of the NZ electricity market, and Section 4 contains the conclusions.

## 2. Nodal, Zonal, and Uniform Pricing Systems: A Short Critical Literature Survey

Access to correctly functioning energy markets in any economy is a key target to boost economic activity. The energy-growth nexus has been widely assessed in the energy economics literature (Ozturk 2010). Empirical research has revealed strong evidence of a possible causal relationship between energy consumption and economic growth, where four different hypotheses have been developed: (i) feedback, (ii) neutrality, (iii) growth and (iv) conservation hypothesis. However, energy transition has recently been seen to go hand in hand with the electrification of energy systems. Therefore, apart from understanding the relationship between energy consumption and economic growth, assessing the behaviour

of electricity consumption is paramount in the energy transition path. Papież et al. (2019) concluded that in countries where RES are well developed, electricity consumption is likely to boost economic growth and vice-versa. Therefore, as long as economic activity grows, a significant rise in electricity demand is expected. In turn, electricity demand has played a prominent role in setting the electricity price. In most electricity markets, the price has been positively influenced by the electricity demand. At the same time, the integration of RES, such as wind and solar PV power, tends to reduce electricity prices, provoking an increase in their volatility (Ketterer 2014; Sapio 2019). Increased electricity price volatility may reduce the reliability and resilience of electricity markets (Maniatis and Milonas 2022). This instability may indeed imply a rise in the risk for investors and a decrease in the willingness of new participants to enter electricity markets. In short, understanding the features of electricity prices and under what conditions electricity is commercialized is crucial to ensure that welfare is maximized, and that economic growth is guaranteed. Therefore, the behaviour of electricity price volatility, and the design of new strategies to reduce it, should be carefully assessed. The right market signals should be sent to market participants so that additional investment capacity will likely proceed. Therefore, the normal operation of economic activity could be ensured.

Given the increasing levels of electricity consumption encouraged by the energy transition, and the increasing penetration of RES in electricity markets (and their peculiar effects on electricity prices), unveiling the best efficient electricity pricing mechanism is critical. In this new era of energy transition, well-developed and resilient electricity markets are crucial for economic growth (Souhir et al. 2019). Therefore, this section discusses the impact of integrating wind and solar PV in electricity markets under different pricing mechanisms. The three main electricity pricing systems used in electricity markets worldwide are the zonal, nodal, and uniform mechanisms. The main difference between these pricing systems lies in the management of transmission constraints and which associated costs are accounted for in setting the equilibrium electricity price. The nodal electricity price (also known as the locational marginal price) represents the point of equilibrium between generation, consumption, and transmission constraints at a given frequency period and node. The zonal electricity price is set on the assumption that there are transmission constraints between price zones, but disregards potential constraints in intra-zonal transmission. Lastly, the uniform equilibrium electricity price is determined by the relationship between electricity generation and consumption. The benefits and challenges of each of the aforementioned pricing mechanisms are briefly described by Weibelzahl (2017). The author concludes that nodal pricing systems yield greater efficiency as congestion constraints are directly reflected in electricity prices, and thus, welfare is expected to be maximised. Nevertheless, due to the high number of electricity prices calculated for each trading period, the complexity of this pricing system may hamper coordination within submarkets (or nodes). Table 1 summarises the studies on the MOE across different electricity markets with both heterogeneous generation mix structures and pricing mechanisms.

The studies presented in Table 1 used econometric approaches, and the MOE is confirmed in all pricing mechanisms; i.e., wind and solar PV power cause downward pressure on wholesale electricity prices. Nevertheless, the electricity produced from these intermittent RES seems to increase the volatility of electricity prices. It should be noted that the volatility of electricity prices was empirically assessed primarily for European electricity markets, where uniform and zonal pricing mechanisms are applied (Ketterer 2014; Cludius et al. 2014; Rintamäki et al. 2017; de Lagarde and Lantz 2018; Benhmad and Percebois 2018; Macedo et al. 2021). The magnitude and impact of the MOE from RES very much depends on the production generation mix of the respective electricity market, and the capacity of its electricity interconnections. For instance, in Sweden—which belongs to the Nord Pool electricity market—electricity produced from wind power decreased wholesale electricity prices, and its magnitude was shown to be stable throughout the day (Macedo et al. 2021). In the Nord Pool electricity market, backup is provided by hydro power, which has low marginal production costs. Therefore, the electricity it produces contributes significantly to

smoothing the MOE. Furthermore, the Nord Pool electricity market is recognised as one of the best integrated electricity markets that is highly capable of sharing surplus electricity production from RES between member countries. Meanwhile, in Spain, the same impact of wind power production on reducing electricity prices was also observed, although its magnitude was found to oscillate during the day (Macedo et al. 2022). In the Spanish electricity system, natural gas is the main backup generation source, whose high marginal cost is principally responsible for these shifts in wholesale electricity prices.

**Table 1.** Summary of the studies focused on the MOE in electricity markets with distinct electricity pricing mechanisms.

| Zonal Electricity Pricing Mechanism | | | | |
|---|---|---|---|---|
| **Authors** | **Period Analysed** | **Electricity Market** | **Method** | **Main Findings** |
| Macedo et al. (2021) | Daily data from January 2016 to April 2020 | Nord Pool Electricity Market (Sweden) | Seasonally Adjusted Autoregressive Moving Average (SARMA)/Generalized Autoregressive Conditional Heteroskedasticity (GARCH) | Wind power ↓ electricity prices, and this impact is similar in magnitude in the 24 h of the day. |
| Maciejowska (2020) | Daily data from January 2015 to January 2018 | German | Quantile Regression Model | Solar and wind power ↓ electricity prices; their prominent impact occurs during peak prices. |
| Sapio (2019) | Daily data from September 2006 to July 2015 | Italian | Quantile Regression Model | Solar and wind power ↓ electricity prices; solar power is likely to ↑ price volatility, more than wind; the establishment of a new cable for electricity transmission ↓ price volatility. |
| Benhmad and Percebois (2018) | Hourly data from January 2012 to December 2015 | German | Seemingly unrelated regression | Solar PV and wind power ↓ electricity prices. The impact is likely to vary throughout the 24 h of the day. |
| Papaioannou et al. (2018) | Daily data from January 2004 to December 2014 | Greek | SARMAX/(E)GARCH | This electricity market does not exhibit asymmetries (i.e., leverage or inverse leverage effect) in volatility of electricity prices. |
| Gürtler and Paulsen (2018) | Hourly data from 2010 to 2016 | German | Fixed Effects Regression with Driscoll-Kraay estimator | Solar PV and wind power ↓ electricity prices; their impact is less prominent between 2013 and 2016, due to a ↓ in fuel prices. Reductions in forecast errors on electricity generation from wind and solar PV would decrease price volatility. |
| de Lagarde and Lantz (2018) | Hourly data from 2014 to 2016 | German | Markov Switching Model | Solar PV and wind power ↓ electricity prices in both regimes, i.e., low and high prices; this negative impact is more pronounced in regimes of high electricity prices. |
| Rintamäki et al. (2017) | Houry data from January 2010 to December 2014 (Denmark); Hourly data from January 2012 to December 2014 (Germany) | German and Danish (Nord Pool Electricity Market) | SARMA | Wind power ↓ electricity prices both in Germany and Denmark. In Denmark, price volatility is lower when wind production increases, while high in Germany; these contrasting results are related to flexible power generation capacity in each country. |

**Table 1.** *Cont.*

| Nodal Electricity Pricing Mechanism | | | | |
|---|---|---|---|---|
| **Authors** | **Period** | **Electricity Market** | **Method** | **Main Findings** |
| Csereklyei et al. (2019) | 30-min and daily data from November 2010 to June 2018 | Australian | Autoregressive distributed lag regression model | Wind power and solar PV ↓ electricity prices-min estimations); increased electricity dispatched by wind power is associated with a lower magnitude of the MOE from solar PV. |
| Paul et al. (2017) | N/A | Australian | Simulation/sensitivity analysis using ANEM model | Wind power ↓ wholesale electricity prices, albeit ↑ retail electricity prices; underinvestment in interconnection capacity is the main argument for wind power not causing a further ↓ in electricity prices. Reliable capacity of interconnections is highly required in nodal pricing systems. |
| Woo et al. (2016) | Hourly data from December 2012 to April 2015 | Californian Independent System Operator | Iterated Seemingly Unrelated Regression | Real-time and day-ahead electricity prices seems to not converge mainly due to day-ahead forecast errors. |
| Forrest and Macgill (2013) | 30-min data from March 2009 to February 2011. | Australian | Tobit Model | Wind power ↓ electricity prices; larger levels of electricity produced from wind power will likely intensify the downward pressure in electricity prices and reduce the incentives for new wind power players to enter the market. |
| Uniform Electricity Pricing Mechanism | | | | |
| **Authors** | **Period** | **Electricity Market** | **Method** | **Main Findings** |
| Macedo et al. (2022) | Daily data from May 2015 to December 2020. | Iberian (Spain) | SARMAX/GARCH | Wind power and solar PV ↓ electricity prices at most times of the day; the magnitude of these impacts varies substantially for each of the 24 h. |
| Macedo et al. (2020) | Daily data from January 2011 to September 2019. | Iberian (Portugal) | SARMAX/GARCH | Wind power and solar PV ↓ electricity prices, while increasing its volatility. A leverage effect was confirmed. |

Notes: ↓ means decrease and ↑ signifies increase.

Overall, the best-suited pricing mechanism to efficiently integrate RES into electricity markets is still under serious debate in the literature. More recently, the restructuring of Indonesia's electricity system to move towards a wholesale market was assessed under zonal, nodal, and uniform pricing systems by Heffron et al. (2022). The authors suggested that the uniform pricing system seemed to provide a more balanced solution to the energy trilemma of security, sustainability, and affordability, as it entailed a higher investment in RES capacity. In fact, the high level of risk aversion often associated with nodal electricity markets may discourage investment in additional RES capacity in systems that use nodal pricing (Ambrosius et al. 2022). Nodal electricity prices are generally more susceptible to spatial volatility in electricity demand, which may increase electricity price unpredictability. However, further research on this matter is required.

Nevertheless, in what concerns the efficient redispatch of electricity and the avoidance of congestion on transmission lines, which are paramount issues for reducing the volatility of electricity prices, nodal pricing systems seem to provide better performance. For instance, the costs associated with congestion are considered in setting nodal electricity prices but not in setting uniform or zonal prices. However, in systems using zonal pricing, electricity redispatch usually occurs, which tends to exacerbate inefficiencies in electricity production (Sarfati et al. 2019). This phenomenon usually gives rise to the increase-decrease game (in-dec game), in which producers oversell electricity in the day-ahead electricity market to benefit from cheaper electricity in the redispatch stage (Holmberg and Lazarczyk 2015). This mechanism means that electricity producers could then profit from arbitrage opportunities.

In-dec game bidding behaviour is rarely observed in nodal pricing systems, as congestion issues are considered in setting electricity prices. In short, nodal pricing systems seem to manage congestion most effectively in electricity markets.

Regarding the empirical assessment of electricity production from RES in nodal electricity prices, Tsai and Eryilmaz (2018) concluded that an increase in electricity produced from wind power is likely to decrease nodal electricity prices in the Electricity Reliability Council of Texas. The magnitude of this impact was shown to be heterogeneous across time and space. In fact, the MOE from wind power is higher in the West Texas region, where wind power has high electricity production levels (Wiser et al. 2017). In this case, transmission constraints also cause a downward effect on electricity prices. In seven regions of the United States, wind and solar electricity production reduce electricity prices (Mills et al. 2021). Even though these intermittent RES reduce the nodal electricity price, the main driver for its decline is the falling price of natural gas. Short-term power markets seem to efficiently integrate RES (more than traditional electricity markets), as they can more easily sustain flexible conventional generation, as noted in the case of the California electricity market (Bushnell and Novan 2018). Notwithstanding, the magnitude of the MOE from wind and solar power has been studied, but a thorough assessment of their impact on electricity price volatility remains scarce. In fact, as already mentioned, volatility increases the unreliability of electricity markets due to the augment of pricing risks. Furthermore, it should also be stressed that studies assessing nodal electricity prices seem to use data files compiled with relatively outdated data, e.g., Woo et al. (2011) and Wen et al. (2022).

Many electricity markets have been restructured and have adopted nodal pricing systems to reduce market failures as, for instance—related to in-dec game issues—including the Californian electricity market (Alaywan et al. 2004). Currently, most electricity markets in the United States use nodal pricing systems. Meanwhile, in Europe, most electricity markets use zonal or uniform pricing systems, although restructuring them to use nodal pricing systems has been discussed in the literature. Lété et al. (2022) discuss in detail the reasons why this transition is not straightforward in Europe and highlight aspirations for an internal European electricity market. To avoid issues such as the in-dec game and arbitrage opportunities, European markets have been advocating a Flow-Based Market Coupling strategy (see, e.g., Van den Bergh et al. 2016; Sarfati et al. 2019). Furthermore, additional storage capacity has been shown to deal effectively with congestion constraints in zonal electricity markets and decrease inter-temporal volatility in electricity prices (Weibelzahl and Märtz 2018).

Although it is fair to say that zonal and uniform pricing systems have an inefficient redispatch mechanism, they still encourage competitiveness and reduce undue market power. In contrast, in nodal pricing systems, market power is likely to be more pronounced and less competitive (Antonopoulos et al. 2020). However, the structure of nodal pricing systems allows the active participation of electricity consumers, and also enables local electricity distribution, such as by implementing a distributed generation strategy.

Choosing an optimal market design to efficiently integrate RES and enable a smooth transition to clean energy is still unclear and needs further research. Although there have been many studies of zonal and uniform pricing systems, or even nodal pricing systems, a detailed empirical assessment of the behaviour of the volatility of nodal electricity prices remains scarce in the literature, mainly because modelling them is so complex. Thus, there is a gap in the literature, but as will be shown, a lack of suitable data remains a major obstacle to filling this gap. Therefore, a principal aim of this paper and its novel contribution to the literature is to specifically identify these obstacles so that usable information on nodal electricity markets can be made available and enable further study on the impact on electricity prices of integrating RES in insular electricity systems with nodal pricing mechanisms.

## 3. Challenges to Assessing the Behaviour of Electricity Prices in NZ

### 3.1. Brief Overview of the Energy Mix in NZ

The electricity market in NZ is an insular power system that has no transactions with other electricity systems and uses a nodal electricity pricing system. In the NZ electricity market, most of the electricity is traded on the wholesale market and electricity prices are established every half-hour for each grid injection or exit point, i.e., for around 250 nodes. Electricity price differences between nodes may occur when there are losses and congestion on the transmission lines.

According to the data published by the Ministry of Business, Innovation & Employment of NZ, hydropower supplied around 56% of total net electricity generation in 2020, followed by geothermal power, which contributed 18%. Meanwhile, the share of electricity generation from oil, gas, and coal was around 0.04%, 14% and 5%, respectively. Most of this electricity is generated by five major electricity companies; Meridian Energy, Genesis Energy, and Mercury Energy, which are state-owned, and Contact Energy and Trustpower, which are private companies. According to the last report published by the Electricity Authority of NZ, these five companies supplied around 89% of the total electricity generation in 2018. Furthermore, these five major electricity companies also operate in the retail electricity market, so there is a high level of vertical integration in NZ's electricity system.

The crude oil produced in NZ is of high quality and almost entirely exported to benefit from its premium price advantage (International Energy Agency 2017). Thus, the country is a net importer of oil, but a net exporter of coal. Meanwhile, natural gas transactions in this country are from indigenous production. Indeed, NZ has a massive resource base, although the contribution of oil, gas, and coal to this country's economy has declined recently. This issue may raise concerns for the electricity sector, as NZ's electricity system is dominated by hydro-power. The installed capacity of non-RES (mainly gas) has proved essential in ensuring the security of the electricity supply during dry years. Fundamentally, the insular NZ electricity system does not depend on imports.

The seasonal weather conditions in NZ increase the vulnerability of its electricity supply. During the winter, when electricity demand is at its highest level, much of the precipitation falls as snow, not rain. All this 'locked-up' water does not flow into hydro-reservoirs, limiting their generation potential and increasing wholesale electricity prices. When the snow starts to melt in the spring, it flows into the hydro-reservoirs and this inflow continues over the summer, unfortunately coinciding with a period of lower electricity demand. Ultimately, hydropower capacity is limited to the volume of water that can be stored. The volume of water in hydro lakes is clearly out of synch with seasonal electricity demand in NZ. Therefore, the full potential of hydropower in the NZ electricity system is underexploited. To reduce possible collateral effects of this mismatch between hydropower production and electricity demand, such as spikes in electricity prices and increased instability in wholesale electricity prices, the NZ electricity system has introduced two main initiatives: a customer compensation scheme, and hedge contracts, supplemented by some ancillary services such as instantaneous reserve, frequency keeping, and/or over-frequency reserve. However, there is insufficient evidence on whether these schemes impact the behaviour of electricity prices.

### 3.2. Challenges in Processing Big Data on Nodal Electricity Markets: The Suitability of Econometric Models

The Electricity Authority of NZ provides the Electricity Market Information (EMI) website with plenty of information on the retail, wholesale, and forward electricity markets. The information published includes several .csv files containing market performance metrics and tools. To optimise data-processing time, the Python language[1] was used for downloading, filtering, and storing data. The data were obtained from the EMI official website, from January 2017 to March 2021, for all nodes (around 250). The variables included in the database are wholesale electricity price (PRICE, $/MWh), electricity production from wind (WIND, MWh) and hydro (HYDRO, MWh), and electricity consumption (CONS, MWh).

Grid injection (INFLOW, MWh) and grid offtake (OUTFLOW, MWh) are used as proxies of the electricity inflow and outflow, respectively. To reduce the feature of non-normality of data, all variables were transformed into their natural logarithms. Please note that this study only considers the electricity produced from wind and hydro power, as the installed capacity of solar PV was insignificant in the NZ electricity market during the period studied. The primary reason for this analysis was to assess the impact of wind power on the volatility and mean price of wholesale electricity in NZ.

The most suitable method to apply in this situation is the Generalized Autoregressive Conditional Heteroskedasticity (GARCH) model of Bollerslev (1986). The GARCH model is an econometric approach widely used in recent literature in the research field of economics (see e.g., Ciarreta and Zarraga 2016; Macedo et al. 2020; Ciarreta et al. 2020) to interpret the behaviour of the volatility persistence of variables with substantial variability in standard deviations. Understanding how electricity pricing volatility is being positively or negatively affected by potential independent variables (which in the case of this research are: wind, electricity consumption, electricity inflow, and outflow), as well as by previous or futures shocks, is crucial to design the optimal strategy for electricity prices' formation. In fact, assessing the behaviour of volatility of electricity price provides valuable information for producers to evaluate market competitiveness and prevent pricing risks (Ioannidis et al. 2021). Finally, a detailed analysis of electricity price volatility is fundamental for policymakers to ensure the correct functioning of electricity markets through the design of appropriate regulations and directives considering specific features of electricity prices' volatility. Moreover, bearing in mind the strong seasonality of electricity prices, the GARCH model was combined with the Seasonal Autoregressive Moving Average model with exogenous regressors (SARMAX) of Box and Jenkins (1976).

Processing the EMI data was computationally demanding, as each .csv file contains information for each node for one day at an hourly, half-hourly, or even 5-min frequency[2]. The setting of nodal electricity prices is not straightforward (compared to the zonal or uniform pricing mechanisms), so assessing the behaviour of electricity prices at each node is challenging and time-consuming. To overcome this issue, we used three different procedures to obtain raw data with as few bugs as possible. Briefly, the three procedures consisted of: (i) collecting the data and compacting it for each island, as the analysis of the MOE from wind power was intended to be analysed separately for the North Island (NI) and South Island (SI); (ii) in the second procedure, because of the misspecification of the econometric models in the first procedure due to a loss of information between POC codes, we analysed the nodes with the highest levels of electricity consumption and production from wind power in both the NI and SI; (iii) lastly, in the third procedure, due to the issue of missing values discovered in the second procedure, the process of data collection for each region was described.

The first procedure aimed to assess the dynamics and behaviour of wholesale electricity prices in both islands of NZ. In the NI, where most of the population lives, there is an electricity production deficit, i.e., electricity consumption is higher than electricity production. In contrast, in the SI, there is an electricity production surplus. This situation was one of the main reasons to study wholesale electricity prices in the NI and SI separately.

The mean of the wholesale electricity price for all nodes in the NI and SI, their respective electricity inflows and outflows, and the means of electricity generation from wind and hydro at 30-min intervals, were calculated. We attempted to estimate the SARMAX/GARCH models using 30-min data for the NI and SI regions. However, the convergence of estimations was not achieved, which is a sign of misspecification and low robustness of the models. Therefore, different techniques were applied to control eventual misspecification issues, such as filtering outliers and testing other flexible distributions to account for the negative skewness and high kurtosis usually observed in electricity price time series (e.g., t-location scale). In addition, seasonality was also analysed in depth using boxplots, as suggested by Ioannidis et al. (2021). However, the SARMAX/GARCH models still appeared to be mis-specified, so the models were re-estimated using hourly data. In

this case, the estimations revealed non-stationarity, as the sum GARCH error parameter and the GARCH lag parameter was higher than 1, as shown in Table 2.

**Table 2.** SARMAX/GARCH models using hourly frequency data.

|  | NI | SI |
|---|---|---|
| **Variables** | **Mean Equation** | |
| LCONS | −0.3199 *** | 1.1861 *** |
| LWIND | −0.0965 *** | −0.0065 *** |
| LHYDRO | 0.5901 *** | −0.1928 *** |
| $\omega$ | 4.1054 *** | −3.5858 *** |
| AR(1) | 0.9753 *** | 0.9770 *** |
| SAR(24) | 0.0865 *** | 0.0883 *** |
| MA(1) | −0.4089 *** | −0.4286 *** |
| SMA(1) | 0.3422 *** | 0.3601 *** |
|  | **Variance Equation** | |
| C | −0.0610 *** | −0.0495 ** |
| $\alpha$ | 1.1627 *** | 1.1764 *** |
| $\beta$ | 0.3882 *** | 0.4388 *** |
| LWIND | −0.0006 | 0.0004 * |
| LCONS | 0.0162 *** | 0.0101 *** |
| LHYDRO | −0.0092 *** | −0.0027 |
|  | **Diagnostic Tests** | |
| $R^2$ | 0.8443 | 0.8524 |
| AIC | −0.5504 | −0.5580 |
| SIC | −0.5455 | −0.5545 |
| $Q_{LB}(24)$ | 451.61 [0.000] | 652.89 [0.000] |
| $Q_{LB}^2(24)$ | 117.35 [0.000] | 212.18 [0.000] |
| ARCH(24) | 4.9840 [0.000] | 9.0092 [0.000] |
| Inverted AR Roots | Stationary | Stationary |
| Inverted MA Roots | Stationary | Stationary |

**Notes:** $Q_{LB}$ stands for the serial correlation teste of standardized residuals; $Q_{LB}^2$ denotes the serial correlation test of the standardized squared residuals; ***, **, * correspond to the 1%, 5% and 10% level of statistical significance, respectively; in [] are presented the *p*-values; AR and SAR terms are the non-seasonal and seasonal autoregressive parameters, respectively; and MA and SMA represent the non-seasonal and seasonal moving average parameters.

Analysing the relationship between electricity pricing and variables at such a high number of nodes filtered into NI and SI data is extremely complex. The data at each node are represented by POC codes which vary over time, so any data glitches may contaminate the estimations. Thus, when misspecification errors arose when running the SARMAX/GARCH models during the first procedure, we suspected they were due to omitted variables or bugs in the database (such as missing information). Subsequent investigation, including the analysis of the corresponding residuals (Figure 1a,b), led us to conclude that this was the case.

The residuals of the hourly data estimations show both downward and upward spikes, which means that independent variables are not explaining well the dependent variable, revealed by the actual residuals. This could be explained by errors in the database and/or the omission of highly significant variables that actually contribute to setting nodal electricity prices. There are two main reasons for this. Firstly, while zonal and/or uniform electricity prices are determined by the equilibrium between electricity produced and electricity consumption, commonly known as marginal markets, the setting of nodal electricity prices also includes the costs associated with transmission losses and congestion on transmission lines. Data on these latter two variables are not available on the EMI website (and it is impossible to calculate a proxy), so this became a limitation for this research. Secondly, in NZ, there is an electricity production surplus from hydropower in the SI, but a deficit in the NI. Therefore, electricity is usually transferred from the SI to

the NI using a high-voltage direct current (HVDC). This flow of electricity through the HVDC link substantially impacts the electricity prices set in the NI and SI. During dry seasons, thermal power (which has high production costs) is transferred from the NI to the SI, so upward price spikes are common, whereas during wet seasons, hydropower (which has low production costs) is transferred from the SI to the NI, making downward price spikes more likely. Variables were introduced in the models to represent the inflow and outflow via the HDVC link between the NI and SI. However, this still did not rectify the low explanatory power of the estimations.

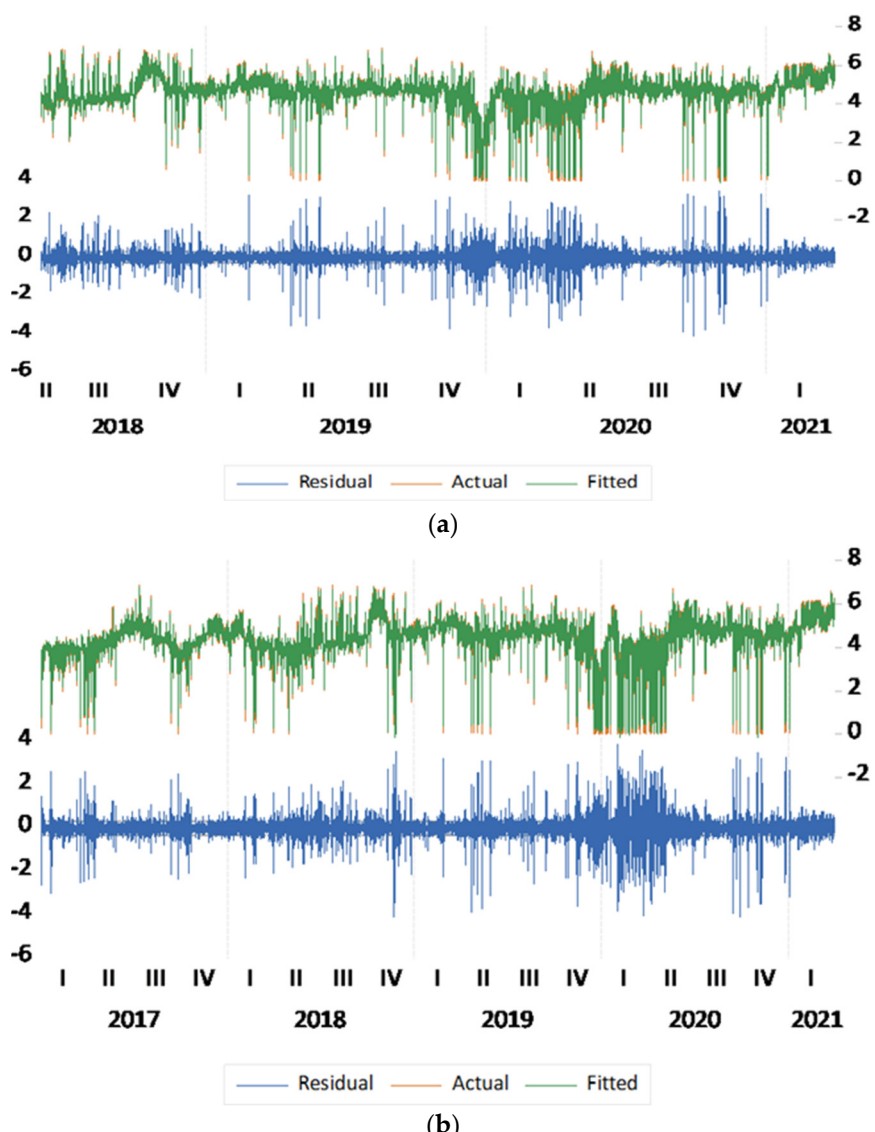

**Figure 1.** (**a**) Residual, Actual, and Fitted Residuals of the hourly SARMAX/GARCH estimation of NI. (**b**) Residual, Actual and Fitted Residuals of the hourly SARMAX/GARCH estimation of SI.

Thus, in this first procedure, we identified two key issues. Firstly, there are crucial variables that should be included in the estimations to explain nodal electricity prices, such as transmission losses and congestion on transmission lines, for which no data are currently available. Secondly, the flow of electricity between the NI and SI via the HDVC link must be considered in estimations regarding the behaviour of New Zealand electricity prices, and these variables have not been considered in recent literature (e.g., Wen et al. (2022)). However, it is currently impossible to make a separate assessment of wholesale electricity prices in the NI and SI because of the lack of information on the POC codes, which change over time. This issue, identified during the first procedure, was also a limitation for the

two subsequent procedures described in this paper, and the challenges of filtering the POC codes led us to formulate the revised structure of the second procedure.

Given that data shortcomings made it impossible to properly analyse electricity prices by node or by island, a second procedure was devised to assess the MOE in nodes with high levels of wind-powered electricity generation and electricity consumption. This criterion was chosen because the core of this research was to examine the MOE from wind power in the NZ electricity market. As wholesale electricity prices in both islands were thought likely to converge due to the high capacity of electricity interconnections, it seemed reasonable to look in detail at nodes in each island that represent the highest levels of consumption and production. This was a tough decision, as nodes with data on electricity consumption did not necessarily have data on electricity generation and vice-versa, and this approach made it difficult to study the reference nodes suggested by the Electricity Authority, namely: Benmore, Haywards, Invercargill, Otahuhu, Stratford, Stoke, and Whakamaru. Therefore, to simplify the choice of nodes, we considered the 19 nodes previously studied by Young et al. (2014), and recently considered by Wen et al. (2022). Nonetheless, even within these 19 nodes, data were still missing for electricity demand and electricity generation.

In the third and last procedure, once more attempting to overcome the matter of missing values, an assessment of the MOE in the NZ electricity system was made by region (e.g., in the Clyde region the POC codes are CYD0331, CYD2201; CYD2201 CYD0). This was performed by using the values of electricity generation, electricity consumption and electricity flow for each POC code. However, when building the database, inconsistencies were still found in the data for a few nodes. These inconsistencies included electricity inflow being identical to electricity generation, or the sum of total electricity produced and electricity inflow being lower than total electricity consumption.

Thus, in all three procedures, fundamental limitations were found. Firstly, the data contained substantial gaps and inconsistencies, so the GARCH estimations for each node produced spurious results and indicated that the models were mis-specified. Secondly, there were no data on certain variables that significantly impact the price of electricity in New Zealand. Therefore, the complexity of the NZ electricity market and the characteristics and limitations of the available data prevented us from properly estimating the GARCH models for each node. In the absence of a proper empirical analysis with robust results, we were unable to reach detailed conclusions for our initial research question. Nonetheless, we were able to pinpoint the exact data that are urgently needed for future studies on the behaviour of nodal pricing mechanisms and that are essential for formulating successful policies. Figure 2 shows a diagram summarising the procedures followed.

In summary, the complexity of modelling nodal electricity pricing systems makes it difficult to undertake academic or technical evaluations of their effectiveness. At a time when these mechanisms are considered as a vital part of the energy transition, a benchmarking analysis of the three most common pricing mechanisms (uniform, zonal, and nodal) is urgently needed to identify the regularities and inefficiencies they produce in electricity markets. To achieve this in the specific case of nodal pricing systems, it is essential that more detailed and reliable data are made available. Despite the fact that it was not possible to complete an empirical assessment, our investigation did give us fresh insight into the behaviour of nodal electricity pricing systems in insular electricity systems incorporating intermittent RES, and these are discussed in the following section.

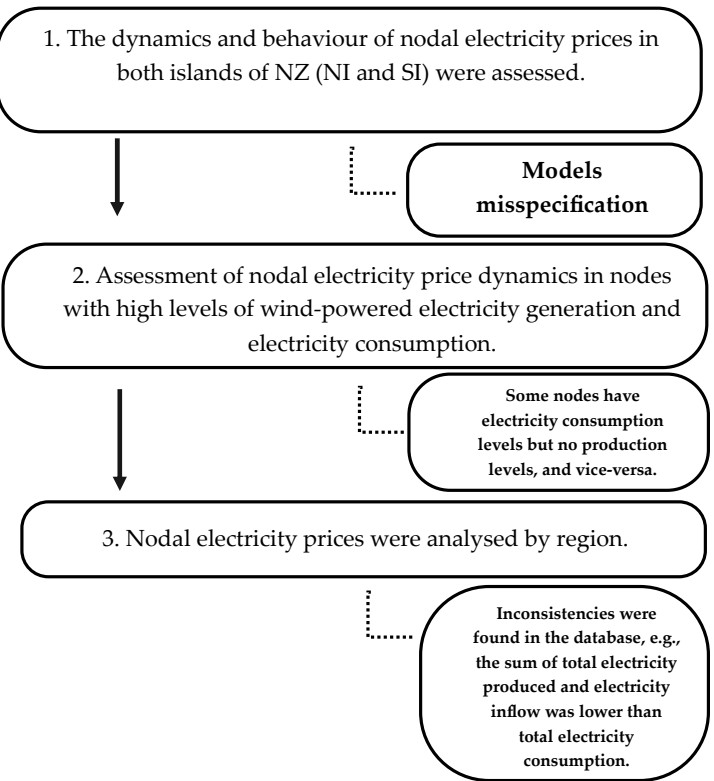

**Figure 2.** Diagram summarising the procedures followed.

## 4. Discussion

Recent increases in electricity production from wind and solar PV power have been incentivised by policies that guarantee prices, such as the feed-in tariffs frequently applied in European electricity markets. These policies have been shown to cause increases in electricity prices, and thus raise household electricity bills (Iimura and Cross 2018). In addition to exacerbating energy poverty issues, as observed by Halkos and Gkampoura (2021), feed-in tariffs seem to discourage further production of wind and solar PV power, as Marques et al. (2019) noted in the case of Spain. Thus, feed-in-tariff policies have actually increased the installed capacity of wind and solar PV, but not the actual generation of electricity, which indicates more idle capacity and economic inefficiency. In addition to highlighting the need for regulatory reform, these findings underline the importance of a detailed analysis of pricing mechanisms to fully understand their implications and ensure that the energy transition is economically sustainable.

Given the importance of RES in moving to cleaner energy, the assessment of the impact of RES on electricity prices in electricity systems is a topical issue. As described above, three main pricing mechanisms have been adopted by electricity systems around the world: zonal, uniform, and nodal. It is obviously important to identify which of these market structures are best suited for integrating renewable energy sources into specific electricity systems. The nodal pricing has been applied in various electricity markets in Latin American and in some parts of the United States. However, to the best of our knowledge, there has been no major empirical study or discussion on the impact of the integration of RES in the volatility of nodal prices.

Insular electricity systems are isolated from other electricity markets, and thus, most of them are dependent on imported fossil fuels. Therefore, these kinds of electricity systems usually suffer from three shortcomings: higher electricity prices, a greater need to ensure a secure electricity supply, and lower reliability. Integrating wind and solar PV power into an electricity system usually lowers electricity prices, but entails greater volatility in electricity prices and generation, making them more unstable and dependent on flexible backup capacity. Furthermore, insular electricity systems cannot establish electricity

interconnections to share surplus electricity from intermittent RES and reduce prices and volatility. As insular electricity systems are inherently less reliable and predictable than interconnected ones, does the introduction of intermittent RES further exacerbate these shortcomings?

The literature lacks empirical studies about the behaviour of the volatility of nodal electricity prices in insular systems during the integration of RES. To fill this gap in the literature, this study aimed to empirically assess the impact of electricity produced from RES, especially wind, on both the mean and volatility of nodal prices of electricity produced from RES in an insular electricity system. The electricity market chosen for this study was that of NZ.

The NZ electricity market is an intriguing case to study, because in addition to being an insular system, it uses a complex nodal pricing mechanism. Furthermore, NZ has a high capacity of installed hydropower, which is particularly valuable for an insular electricity system, as it can play a backup role and reduce volatility in electricity prices. Pumped storage facilities have several advantages, as they can: (i) overcome seasonal mismatches between electricity production from hydropower and electricity consumption, and enable electricity demand and supply to be efficiently balanced; (ii) minimise upward price spikes and reduce intertemporal volatility in electricity prices (Ambrosius et al. 2020); (iii) minimise dependence on imported fossil fuels, and increase the autonomy of insular electricity systems (Papadopoulos 2020); and (iv) increase overall welfare. Pumped hydropower storage capacity in the NZ electricity system is somewhat low, given the overall installed capacity of hydropower, so storage strategies should be revised to improve the reliability of its electricity market.

New Zealand's insular electricity system has no interconnections with other electricity systems, and no subsidies are given to RES producers. Despite these policies, wind and solar PV power have grown within its electricity market. According to data provided by the Ministry of Business, Innovation & Employment of NZ, between 2017 and 2020, electricity generation from wind power increased by 10.2%, and that from solar power by 107.4%. These increases suggest that other factors are driving the penetration of RES in its electricity system and raise the question of what role its pricing system plays in this. Understanding the relationship between RES and the nodal pricing system in NZ may provide valuable insights for formulating optimal and strategically efficient pricing systems that promote the smooth integration of RES.

Figure 3 shows the annual electricity generation from wind and solar power in GWh.

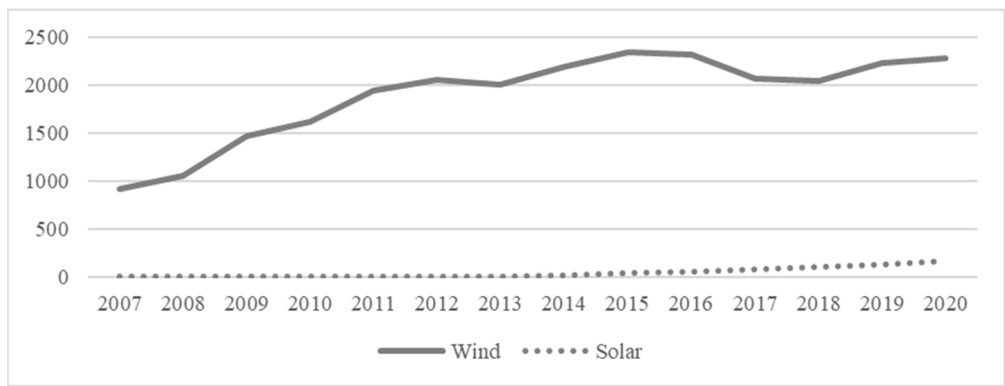

**Figure 3.** Annual electricity generation from wind and solar power in GWh from 2007 until 2020 in NZ. Data source: Ministry of Business, Innovation & Employment of NZ.

In contrast to zonal electricity pricing, nodal pricing considers transmission constraints such as losses and congestion of transmission lines, and thus, more accurately reflects the actual costs involved. The greater transparency this mechanism provides to electricity markets may be one of the reasons that RES have been so readily absorbed by the system in NZ, despite the lack of incentives or additional backup capacity. Verifying the extent to

which nodal pricing is mitigating the potentially adverse effects of RES penetration requires a complex and computationally-demanding empirical analysis of the relationship between prices and RES production. To achieve robust results from this analysis, detailed and accurate data must be made available. This would include data on the costs associated with transmission losses and congestion on transmission lines, and must exclude inconsistent or incomplete data.

The most recent study of the behaviour of nodal electricity pricing in NZ is by Wen et al. (2022), but it only looks at a brief period of somewhat outdated data for 2011 and 2012. Therefore, the current study considered a longer period of more recent data for NZ, from 2017 to 2021. However, as discussed in Section 3.2, the patchy and flawed nature of the data for NZ proved inadequate for estimating the models needed to analyse complex nodal pricing systems, and consequently, the results were not robust and indicated that the models were not well specified (see Table 2. This precluded us from drawing more detailed conclusions (mainly regarding the volatility of nodal electricity prices) or carrying out a benchmarking analysis of the strengths and weaknesses of the three most common pricing mechanisms. Nonetheless, the investigation enabled us to identify some important aspects of the NZ electricity market.

If better data were available and a more robust econometric analysis of the behaviour of electricity prices in NZ became feasible, our research suggests it would find that the volatility of electricity prices induced by intermittent wind generation would be mitigated by the abundant flexible hydro power production. For instance, this flexibility should allow the system to reduce the number of episodes of market failures. Furthermore, as electricity produced from intermittent RES has marginal costs close to zero, it would be expected to reduce electricity prices. Given that RES producers in the NZ electricity system do not have priority access to the grid or receive any incentives, RES investors may receive a lower return on their investment, and this may be slowing the deployment of solar PV and wind power compared to European electricity markets (where uniform and zonal electricity prices is applied). To implement a sustainable transition to cleaner energy, policies should be reformulated to promote greater competitiveness in the market and encourage the deployment of RES.

Lastly, in the NZ electricity market, there is substantial evidence of the presence of market power (Poletti 2021) among the players. Undue market power and entrance barriers may be triggering two interrelated scenarios. Firstly, the lack of competition in the NZ market may be discouraging the development of RES capacity. Secondly, so could heightening inequalities in electricity prices between (or even within) the North and South islands, especially during periods when there is a shortage of electricity production from hydropower. NZ policymakers may be tempted to consider setting a price cap.

## 5. Final Remarks

The optimal generation mix and ideal pricing strategies for a smooth energy transition, while ensuring the maintenance of economic activity are still matters of heated debate in the literature on the subject. The electricity produced from RES is intermittent and weather dependent. As electricity storage capacity is still negligible, electricity markets must become significantly more flexible. However, the first-best strategy to achieve this flexibility is still unclear. In trying to meet ambitious climate change targets, policymakers have been disregarding the role that fossil fuels can still play in electricity systems. In particular, natural gas can be an important transitional energy source, as a recent study by Gürsan and de Gooyert (2021) argues. In addition to this, and as discussed in the literature review, more efficient pricing mechanisms that minimise market failures and incentivise competitiveness are also crucial for a smooth energy transition. The impact of incorporating intermittent RES into electricity systems on these pricing mechanisms needs to be carefully assessed. However, among the most common mechanisms (nodal, zonal, and regional), nodal pricing has not undergone a thorough empirical analysis regarding its volatility. Hence the need for a benchmarking analysis to reveal which of the pricing mechanisms

sends the right price signals to market participants to promote greater integration by RES and enable a sustainable energy transition.

It was noted in the literature review that the impact on zonal and uniform electricity prices of electricity produced from wind and solar PV power has been widely assessed, particularly for European electricity markets. However, there was a distinct lack of empirical studies on the relationship between electricity produced from intermittent RES and nodal electricity prices, namely by deeply assessing their volatility. Thus, the main aim of this study was to better understand why processing nodal electricity pricing data has been so challenging. Among the likely reasons found for the lack of empirical studies is the complexity of setting nodal prices and the computational demands of manipulating the available datasets. This paper's contribution to the field is to alert stakeholders of the urgent need to provide data related to nodal electricity systems that is easier to manage, thereby facilitating the evaluation of the effectiveness of nodal pricing mechanisms in integrating intermittent RES in the transition to cleaner energy. Furthermore, it also examines the primary characteristics and limitations of nodal electricity prices.

The study of insular systems is particularly useful for understanding the feasibility of storage strategies in electricity markets. Furthermore, NZ's pricing mechanism and the electricity market structure mimics the distributed generation strategy. In NZ, the electricity produced in small-scale power plants is usually injected into local distribution networks. A lack of electricity data precluded a deeper and more robust assessment of the dynamics of this electricity market under the nodal pricing mechanism, but the insights gleaned from this study on such a topical subject are still of great value to the literature. Many questions still remain regarding the dynamics of the NZ electricity market that deserve further research. These include studies of resource adequacy to manage any constraint on electricity supplies, the suitability of the market design to the generation *mix*, and whether market price signals increase investment in additional RES capacity.

The NZ electricity market is a largely decarbonized electricity market with substantial installed hydropower and geothermal capacity. However, the capacity installed of wind power compared, for instance, to that in European electricity markets, is low. There are three possible explanations for the slow rate of deployment of RES in NZ. Firstly, the nodal pricing mechanism may be sending biased signals to RES producers, and thus increasing market failures. Secondly, the NZ government provides no incentives for RES development. Considering the low return of investments in RES, due to the nearly zero marginal costs of the electricity produced, additional investments in RES capacity could be being discouraged. Investments in RES generation capacity have high start-up and/or fixed costs. This type of barrier may be dissuading new participants from entering the NZ electricity market. Lastly, the electricity systems in NZ reveal high levels of market power. However, it should be highlighted that all three issues need further investigation so that more accurate conclusions and policy implications can be formulated.

In summary, strategies aimed at further integrating wind power into the NZ electricity market should be reviewed, and the use of other investment signals to encourage wind power production should be discussed. Considering the reliance of NZ electricity on hydropower production and given ongoing climate change issues (e.g., periods of drought), policymakers should be aware that the long-term availability of hydropower is not assured. One limitation of this study, the difficulty of making an empirical assessment of nodal electricity prices, prompted one of its main findings: more easily treatable data on nodal pricing systems are essential for evaluating them. Future research should also focus on the impact of inhibiting market power and providing a greater degree of market competitiveness in nodal pricing systems.

**Author Contributions:** Conceptualization: D.P.M., A.C.M., and O.D.; Methodology: D.P.M. and O.D.; Software: D.P.M. and O.D.; Formal analysis: D.P.M.; Investigation: D.P.M.; Resources: D.P.M.; Validation: A.C.M.; Data Curation: O.D.; Supervision: A.C.M.; Project administration: A.C.M.; Funding acquisition: A.C.M.; Visualization: D.P.M.; Writing—original draft: D.P.M.; Writing—review and editing: A.C.M. All authors have read and agreed to the published version of the manuscript.

**Funding:** This research was funded by NECE-UBI—Research Unit in Business Science and Economics, Project No. UIBD/04630/2020 and the grant reference No. DFA/BD/8457/2020, both sponsored by the Portuguese Foundation for the Development of Science and Technology, and of the LUE (Lorraine Université d'Excellence) UHLYS Impact project.

**Institutional Review Board Statement:** Not applicable.

**Informed Consent Statement:** Not applicable.

**Data Availability Statement:** Data can be found at the official website of the Electricity Authority of the New Zealand, in the following link "https://www.emi.ea.govt.nz/Wholesale/Datasets".

**Conflicts of Interest:** The authors declare no conflict of interest.

## Notes

[1] The Python packages used to download, filter, and store data are the following: *pandas*, *numpy*, and *datetime*.
[2] To obtain the database, each *.csv* file was downloaded (i.e., each day was individually downloaded), and then all days were compacted and stored in Python.

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
