# Peer review of "Challenges in Assessing the Behaviour of Nodal Electricity Prices in Insular Electricity Markets: The Case of New Zealand"

_economies, doi:10.3390/economies11060159_

Round 1
Reviewer 1 Report
This paper discusses various issues of nodal prices in insular electricity systems, using the New Zealand electricity market as a case study. Overall, the paper is written in such a way that the issues in the electricity market are understandable, however, there seems to be little relationship between that discussion and the empirical analysis conducted in this paper. The results presented in Table 1 and Fig. 1.a-b are merely estimation results and residuals from the SARMAX/GARCH model, and the price characteristics read from these results are the result of analysis specific to the NZ market environment, which seems to be a separate discussion from the problems with nodal prices. It also seems unreasonable to link the model's failure to converge with the data sample used to the issues with nodal prices (such as the database not being clean). Overall, I think that in order to claim an academic contribution, there needs to be a compelling different analysis to support it.
Also, the model itself is unclear. It would be necessary to be more specific about the model equation and the variables used in it.
Author Response
Response to Reviewer #1
Dear Professor,
We would like to express our sincere thanks for your comments on our paper. We have split your review into several parts to provide a detailed response to each of your comments and suggestions.
Please find below our responses to your queries and suggestions.
Thank you.
Yours sincerely,
Reviewer Comments (RC)
RC1: “(…) there seems to be little relationship between that discussion and the empirical analysis conducted in this paper.”
(…)
“(…) the price characteristics read from these results are the result of analysis specific to the NZ market environment, which seems to be a separate discussion from the problems with nodal prices. It also seems unreasonable to link the model's failure to converge with the data sample used to the issues with nodal prices (such as the database not being clean).”
Reply: We understand your concern regarding the relationship between the empirical analysis conducted and the discussion section. The failure of the econometric procedure to produce robust results meant that these results could no longer be the main focus of the paper. It might be argued that the inclusion of the Generalised Autoregressive Conditional Heteroskedasticity (GARCH) models, mainly served to highlight the challenges of download, manipulate and filter the data published by the Electricity Authority of New Zealand. However, the importance of the paper’s main question is still meaningful, and the general approach followed remains valid. Furthermore, the subsequent procedures that confirmed the flawed nature of the data processing are also instructive for the literature on the subject.
In the NZ electricity market, there are around 250 nodes, and data for each node is produced for every 5-minute period of every day. Therefore, assessing electricity price behaviour at each node is extremely challenging and time-consuming. To overcome this issue, we first attempted to make a separate assessment for the North and South Islands, of the impact on wholesale electricity prices of electricity generation, electricity consumption, and electricity flow.
Aggregated data is generally easy to manage and, although computationally demanding in this case, its computational analysis is normally viable. Heterogeneity was not a concern in this case, as the patterns of electricity consumption and production varied little within each island. However, the results of estimating the GARCH models, presented in Table 1, indicated that the models were mis-specified, so their results could not be considered robust and properly discussed. Nonetheless, with better data, these models could still provide a template for subsequent successful estimations.
We suspected that these misspecifications may have been due to errors or omissions in the database. Therefore, we conducted two further procedures (each of which is described in detail in subsection 3.2) to compact the data and make the econometric analysis more viable. However, during these procedures, more irregularities were found in the database, such as a number of incomplete datasets (which compromised the estimation of GARCH models, if only one node was selected). Furthermore, it was found that no data was available on several crucial variables used to set nodal electricity prices, such as electricity transmission losses, congestion in transmission lines and electricity flow through high-voltage direct current links. Subsection 3.2 provides a detailed description of this and discusses the challenges, faced in processing big data on electricity, with a particular focus on the failures and flaws encountered in processing data on the NZ electricity market.
In the revised debate section, the specific characteristics of insular electricity systems and the challenges they face in incorporating RES are described. We stress the importance of choosing the most appropriate pricing system to facilitate the integration of RES in insular systems, and the need for a benchmark comparison of the three most common electricity pricing mechanisms. The scarcity in the literature of empirical assessments of insular electricity systems that uses nodal pricing is also noted This is a concern as there are several electricity markets where this pricing mechanism is applied; to the best of our knowledge, there is no large discussion on the empirical evidence on the effectiveness of the nodal pricing mechanism in the energy transition. Thank you for prompting us to clarify this.
RC2: “(…) Overall, I think that in order to claim an academic contribution, there needs to be a compelling different analysis to support it.”
Reply: As mentioned in reply to RC1, this research discusses the challenges of processing data in nodal electricity systems. The flaws in the GARCH estimations presented in the paper nonetheless provide a valuable contribution to the literature by illustrating the vital importance of complete and accurate data and, specifically, identify certain variables for which data must be made available for proper and much-needed assessments of nodal systems to be made. This contribution of the research to the literature was rewritten to clarify this issue. Thank you for your comment.
Reviewer 2 Report
This paper debates the challenges and difficulties faced when assessing nodal electricity price formation. Overall, this study makes contributions to the current literature. I am overall positive regarding the work, which is why I would be open to reviewing a revised manuscript; however the revisions are major. Below presented some detailed comments. Hopefully they can assist the authors to improve the quality of the paper.
1. The Introduction should further motivate the study. Why this study is necessary? What policy level problem this study is addressing? How the study is expected to provide any solution to that problem? Try to find out the policy level problem, as academic literature will not be able to provide you with the specific policy issue.
2. In the introduction and literature review, you need to discuss the existing literature to improve the motivation of the paper and broaden the view of readers and display the contribution of this paper.
3. What is the aim of the review of literature? The authors have merely listed out the studies without even creating a debate among them. Without that debate and thoughtful contradictions, the research gap cannot be substantiated.
4. You need to provide more analysis about the economic reasoning.
5. Conclusion – Please outline a summary of findings, contributions, implications, limitations and avenues for future research. Especially, expand the discussions relating to implications, limitations and avenues for future research.
6. Finally, this manuscript needs careful editing by someone with expertise in technical English editing paying particular attention to sentence structure so that the goals and results of the study are clear to the readers. Please there are considerable number of typos, spelling errors and grammatical mistakes throughout the paper that a careful reading will help you to eliminate them.
Author Response
Response to Reviewer #2
Dear Professor,
We would like to thank you for your suggestion and comments on our paper. We are particularly grateful for the opportunity to clarify the main goal of the manuscript, strengthen the literature review and, summarise the main findings and contribution of this paper in the conclusion section.
Please find enclosed our responses to your comments.
Thank you.
Yours sincerely,
Reviewer Comments (RC)
This paper debates the challenges and difficulties faced when assessing nodal electricity price formation. Overall, this study makes contributions to the current literature. I am overall positive regarding the work, which is why I would be open to reviewing a revised manuscript; however the revisions are major. Below presented some detailed comments. Hopefully they can assist the authors to improve the quality of the paper.
RC1: The Introduction should further motivate the study. Why this study is necessary? What policy level problem this study is addressing? How the study is expected to provide any solution to that problem? Try to find out the policy level problem, as academic literature will not be able to provide you with the specific policy issue.
Reply: Following your suggestions, and as mentioned in reply to RC2, in the revised introduction section the paper’s motivation was explained more fully, and the problem in terms of policy was clarified. In addition, although admitting that the econometric estimations in our study were inconclusive, we emphasise that a proper understanding of the behaviour of nodal pricing mechanisms in insular electricity markets remains essential for energy policy in New Zealand and elsewhere. Furthermore, we indicate the improvements and additional data needed to achieve such an analysis. Thank you for your valuable suggestions.
RC2: In the introduction and literature review, you need to discuss the existing literature to improve the motivation of the paper and broaden the view of readers and display the contribution of this paper.
Reply: We welcomed your comment which made us realise that the literature review section needed further discussion. As mentioned in reply to RC3, one of the main goals of the literature review was to provide a broad overview of the nodal, zonal and uniform pricing mechanisms. In response to your comments we have expanded this section of the paper. In reviewing the literature on the subject, we found many studies on zonal and uniform pricing but very few which empirically assessed the behaviour of nodal electricity prices. As we now explain in the revised paper, this motivated us to carry out an empirical assessment of a nodal pricing mechanism, and we chose the electricity system of New Zealand (NZ) because, it uses nodal pricing and, as an insular electricity system, has no interconnections with other electricity markets.
It is generally acknowledged that nodal pricing mechanisms are extremely complex and modelling them is computationally demanding. This complexity may well have discouraged many researchers from conducting empirical analyses of electricity systems with nodal electricity pricing mechanisms. Nonetheless, given the importance to energy policy of assessing the behaviour of such system, we decided to proceed with this analysis. In processing the available information on the NZ electricity market, we faced continual challenges due to flawed data and missing variables. Once we suspected that the data was hard to manage to produce robust results, we conducted several procedures on the data to make it more reliable, and these are described in detail in section of 3.2 of the paper.
The main contribution of this paper is twofold. Firstly, it discusses the characteristics of the three main electricity pricing mechanisms, and emphasises the global need for a benchmarking analysis of the relative strengths and weaknesses of all three. It then establishes the lack of empirical evidence on nodal electricity pricing and the contribution the paper can make by focusing on this particular mechanism. Secondly, given the challenges faced in processing the available data, the paper provides a detailed description of the different procedures conducted to obtain data with as few bugs as possible. Although we were unable to successfully conclude an econometric procedure, the paper makes a valuable contribution by identifying key variables in nodal electricity pricing for which data is not yet available and the importance of improving the accuracy of all data.
In short, we trust we have clarified the main contribution of this paper in both the introduction and literature review sections of the revised paper. Thank you.
RC3: What is the aim of the review of literature? The authors have merely listed out the studies without even creating a debate among them. Without that debate and thoughtful contradictions, the research gap cannot be substantiated.
Reply: The aim of the literature review section is to debate the main features of the electricity pricing mechanisms most commonly used worldwide. We thank you for pointing out that this needed further debate, and in the revised paper, have expanded the discussion of the literature. This now considers the potential role of these pricing mechanisms in facing the challenges of energy transition and the associated strategies. The literature review now also contains a short summary of studies on the merit-order effect (MOE); the downward effect on prices of electricity produced from RES, especially wind and solar photovoltaic (PV). Thank you for your insightful comments.
RC4: You need to provide more analysis about the economic reasoning.
Reply: The economic reasoning behind the main aim of this paper was further elaborated in the discussion section. Thank you for your suggestion.
RC5: Conclusion – Please outline a summary of findings, contributions, implications, limitations and avenues for future research. Especially, expand the discussions relating to implications, limitations and avenues for future research.
Reply: Following your suggestion, a summary of findings, contributions and limitations, as well as future research is now included in the conclusion section. Thank you.
RC6: Finally, this manuscript needs careful editing by someone with expertise in technical English editing paying particular attention to sentence structure so that the goals and results of the study are clear to the readers. Please there are considerable number of typos, spelling errors and grammatical mistakes throughout the paper that a careful reading will help you to eliminate them.
Reply: The revised manuscript was sent to an English professional reviewer. We hope that the manuscript is now much more reader friendly. Thank you for your suggestion.
Reviewer 3 Report
This paper mainly studies the evaluation barriers of nodal electricity prices in the New Zealand electricity market and its correlation with the impact of renewable energy such as wind and photovoltaic. I think this paper is interesting. I thank the authors. However, the manuscript does not meet the publishing requirements at present, and the following key problems need to be solved.
Comment 1: Some abbreviations appear in their full names only in the body part. In fact, abbreviations should be written in full when they first appear in the abstract and body of the paper.
Comment 2: There is too much introduction to the research background in the abstract, and there is no key description of the problems to be solved, solutions and conclusions of this study.
Comment 3: The serial number of the chapter title is chaotic and needs to be readjusted.
Comment 4: It is suggested to summarize the existing research problems and corresponding solutions after the literature review, and highlight the work and contributions of this research.
Comment 5: At the beginning and at the end of the article, the authors talked about the relationship between nodal electricity prices and wind-photovoltaic renewable energy, but in the middle of the study only mentioned wind resources. It seems that it is lack of strict logic to give a general view on the development of wind-photovoltaic renewable energy in New Zealand's electricity market at the end.
Comment 6: The paper points out the limitations of many existing studies. Although some empirical analysis is given for some problems, there are still errors. The reason for this error is attributed to data problems. It is suggested that the author find a reasonable solution to strongly demonstrate the following conclusions.
Comment 7: The problems and corresponding measures mentioned in the last discussion can be generally recognized at present, and cannot be closely linked with the previous empirical analysis. The whole study lacks objective reasoning and the logic of discussion is not rigorous enough.
Comment 8: There are too few chart elements in the paper, so it is suggested that the author can organize the research ideas into a frame chart to show. Other argumentations with strict logic in the article can also be shown with pictures to enhance the readability of the article.
Author Response
Response to Reviewer #3
Dear Professor,
We would like to express our sincere thanks for your suggestions for improving the paper. We have carefully considered your comments on the reasoning behind the econometric issues presented and clarified it in the revised paper. Furthermore, the abstract and background of the main goal of this research were also revised.
Please find enclosed our responses to your queries and comments.
Our best thanks,
Yours sincerely,
Reviewer Comment (RC)
This paper mainly studies the evaluation barriers of nodal electricity prices in the New Zealand electricity market and its correlation with the impact of renewable energy such as wind and photovoltaic. I think this paper is interesting. I thank the authors. However, the manuscript does not meet the publishing requirements at present, and the following key problems need to be solved.
RC1: Some abbreviations appear in their full names only in the body part. In fact, abbreviations should be written in full when they first appear in the abstract and body of the paper.
Reply: Please accept our apologies for this. The convention on abbreviations has now been made consistent in the body of the paper. Thank you for your alert.
RC2: There is too much introduction to the research background in the abstract, and there is no key description of the problems to be solved, solutions and conclusions of this study.
Reply: We accept your valid criticism. As a consequence, we have heavily revised the abstract and the key description of the problems to be solved, as well as the solutions and conclusions of the study. Thank you.
RC3: The serial number of the chapter title is chaotic and needs to be readjusted.
Reply: In response to this comment we have checked and adjusted the numbering of the sections in the revised paper. Please advise us if this is not what you mean by ‘chapter title’. Thank you.
RC4: It is suggested to summarize the existing research problems and corresponding solutions after the literature review, and highlight the work and contributions of this research.
Reply: Inspired by your suggestion, at the end of the literature review we have introduced a paragraph which identifies the gap we identified in existing research, as well as the main contribution and novelty of our research. Thank you for your suggestion.
RC5: At the beginning and at the end of the article, the authors talked about the relationship between nodal electricity prices and wind-photovoltaic renewable energy, but in the middle of the study only mentioned wind resources. It seems that it is lack of strict logic to give a general view on the development of wind-photovoltaic renewable energy in New Zealand's electricity market at the end.
Reply: Thank you for your comment. Policies worldwide are targeting the massive integration into electricity markets of Renewable Energy Sources (RES), especially wind and solar photovoltaic (PV). Therefore, in the introduction, debate and conclusion section we discuss the effects on nodal electricity pricing of integrating these two emerging technologies. However, in Section 3, we only consider the electricity produced from wind power, as the capacity installed of solar PV power still insignificant in the New Zealand (NZ) electricity market. We recognise that this issue needed to be clarified, and have done so in subsection 3.1 of the revised paper. Thank you suggesting this improvement.
RC6: The paper points out the limitations of many existing studies. Although some empirical analysis is given for some problems, there are still errors. The reason for this error is attributed to data problems. It is suggested that the author find a reasonable solution to strongly demonstrate the following conclusions.
Reply: We are afraid we may not have fully understood this comment, so if our response does not properly address it, please let us know. The NZ electricity market is an insular electricity system with no electricity interconnections to other electricity markets, and which uses a nodal pricing mechanism. The Electricity Market Information website, provided by the Electricity Authority of NZ, has plenty of information considering the retail, wholesale and forward electricity markets. Each file contains information for one day and consists of data with an hourly, half-hourly, or even 5-min frequency. Therefore, assessing the nodal pricing mechanism is complex and computationally demanding. The Python language was used for downloading, filtering and storing the data. To study the behaviour of the NZ electricity market, and the Merit-Order Effect (MOE), should include as many dynamics in electricity markets as possible. However, studying each node individually, besides being time consuming, it is not really feasible as, if data shows electricity is being consumed at a node, it does not necessarily mean that the electricity is being generated there, and vice-versa.
To overcome these issues, several different procedures were followed to improve the reliability of the data, and these are described in detail in section 3.2 of the paper. Briefly, to make the econometric analysis feasible, we decided to aggregate all the electricity production and electricity consumption data by island, dividing it into North Island (NI) and South Island (SI) data. Aggregated data is easier to manage and more computationally feasible to analyse. Heterogeneity was also not a concern, as the patterns of electricity production and consumption in each island are much the same. Nonetheless, the results produced by this analysis indicated that the models were mis-specified, and the signs of certain results were unreasonable, such as the negative impact of electricity consumption on electricity prices. Furthermore, it was evident in the graphs of residuals that the dependent variable was not well explained by the independent variables. There are two likely explanations for this. Firstly, a loss of information on electricity production or consumption variables during the filtering of the data. This type of loss of information is hard to circumvent, as the gaps and mismatches in series change node by node, making selecting the optimal criterion for filtering data very challenging. Secondly, there are several variables considered in setting nodal electricity prices that are not published. These include electricity transmission losses, congestion in transmission lines and the electricity flow between the NI and SI through the high-voltage direct current link between them.
Despite these limitations, two more procedures were followed: (i) a second procedure, addressed the misspecification of the econometric models in the first procedure, and analysed nodes in NI and SI which had the highest levels of electricity consumption and wind powered production; and (ii) a third procedure, which addressed the missing values discovered in the second procedure, processed the data collected for each region. While processing the data, many inconsistencies were found in the database, such as discrepancies between files (e.g., the sum of total electricity produced and electricity inflow was lower than total electricity consumption in few a nodes, and there were several incomplete files making estimation of GARCH models unfeasible. In view of all this evidence, we concluded that the misspecification of the models could indeed be related to data issues. In the revised paper we have clarified how we concluded that the misspecification of the empirical analysis models was related to data issues. Thank you for prompting us to explain this.
RC7: The problems and corresponding measures mentioned in the last discussion can be generally recognized at present, and cannot be closely linked with the previous empirical analysis. The whole study lacks objective reasoning and the logic of discussion is not rigorous enough.
Reply: Spurred by this observation, we have strengthened the ideas and arguments in the revised discussion and final remarks sections. Furthermore, in the introduction section we have also clarified the main goal of the study. Thank you for your comment and please let us know if these changes do not fully address it.
RC8: There are too few chart elements in the paper, so it is suggested that the author can organize the research ideas into a frame chart to show. Other argumentations with strict logic in the article can also be shown with pictures to enhance the readability of the article.
Reply: We have included a graph in subsection 3.2 summarising the various procedures used to manipulate the nodal electricity pricing data. We hope the paper is now more reader friendly. Thank you for your suggestions.
Round 2
Reviewer 1 Report
I confirm that certain explanations have been reinforced in response to the previous points raised. I appreciate the authors' hard work to improve their research.
However, with this revision, the claim is emphasized in various places that there are very few empirical studies on nodal prices, which I believe is not correct. In fact, a very large number of papers in the U.S. have empirically analyzed the impact of RES on nodal prices.
Examples are given below, but in addition to them, several other studies, such as those described in Table 1 of Mills et al. (2021), for example, have conducted similar studies.
It seems necessary to survey these studies appropriately and to discuss more clearly what analysis is required based on the results of these studies, and what contribution this study makes to these issues.
- Mills, A., Wiser, R., Millstein, D., Carvallo, J. P., Gorman, W., Seel, J., & Jeong, S. (2021). The impact of wind, solar, and other factors on the decline in wholesale power prices in the United States. Applied Energy, 283, 116266.
- Bushnell, J., & Novan, K. (2018). Setting with the sun: The impacts of renewable energy on wholesale power markets (No. w24980). National Bureau of Economic Research.
- Wiser, R. H., Mills, A., Seel, J., Levin, T., & Botterud, A. (2017). Impacts of variable renewable energy on bulk power system assets, pricing, and costs. Lawrence Berkeley National Lab.(LBNL), Berkeley, CA (United States).
- Tsai, C. H., & Eryilmaz, D. (2018). Effect of wind generation on ERCOT nodal prices. Energy Economics, 76, 21-33.
Minor points:
- It would be better to have a short description of some of the Variables in Table 1.
- The graph in Figure 1 should be enlarged horizontally a little more to make it easier to read.
Author Response
Dear Professor,
Thank you, once again, for your careful revision of our paper. All your suggestions have been carefully considered and incorporated into the revised paper. We are particularly grateful for the opportunity to enhance the discussion about the integration of wind and solar power in nodal electricity markets. Please find enclosed our responses to your queries and suggestions.
Thank you.
Yours sincerely,
Reviewer Comment (RC)
RC1: I confirm that certain explanations have been reinforced in response to the previous points raised. I appreciate the authors' hard work to improve their research.
However, with this revision, the claim is emphasized in various places that there are very few empirical studies on nodal prices, which I believe is not correct. In fact, a very large number of papers in the U.S. have empirically analyzed the impact of RES on nodal prices.
Examples are given below, but in addition to them, several other studies, such as those described in Table 1 of Mills et al. (2021), for example, have conducted similar studies.
It seems necessary to survey these studies appropriately and to discuss more clearly what analysis is required based on the results of these studies, and what contribution this study makes to these issues.
- Mills, A., Wiser, R., Millstein, D., Carvallo, J. P., Gorman, W., Seel, J., & Jeong, S. (2021). The impact of wind, solar, and other factors on the decline in wholesale power prices in the United States. Applied Energy, 283, 116266.
- Bushnell, J., & Novan, K. (2018). Setting with the sun: The impacts of renewable energy on wholesale power markets (No. w24980). National Bureau of Economic Research.
- Wiser, R. H., Mills, A., Seel, J., Levin, T., & Botterud, A. (2017). Impacts of variable renewable energy on bulk power system assets, pricing, and costs. Lawrence Berkeley National Lab.(LBNL), Berkeley, CA (United States).
- Tsai, C. H., & Eryilmaz, D. (2018). Effect of wind generation on ERCOT nodal prices. Energy Economics, 76, 21-33.
Reply: Thank you, once again, for your careful and detailed revision of our paper. We have rectified this point in the paper. A discussion about the impact of the integration of renewable energy sources, namely wind and solar power, in nodal electricity markets was incorporated in Section 2 of the revised paper by citing the articles suggested by you.
RC2: Minor points:
- It would be better to have a short description of some of the Variables in Table 1.
- The graph in Figure 1 should be enlarged horizontally a little more to make it easier to read.
Reply: Thank you for your sharp insights. A concise description of the variables in Table 1 has been included in its subtitle in the revised paper. Furthermore, the information regarding transforming variables into their natural logarithms has been included in the text of Subsection 3.2. Considering the size of Figure 1, it has been also horizontally enlarged to make it easier to analyse. Overall, we are profoundly grateful for your revision of our paper. We are indebted.
Round 3
Reviewer 1 Report
I reviewed the revisions made to Section 2 and appreciate the changes the authors have made. In regards to Section 4, there are still areas that need to be addressed based on my previous comment, and I recommend that the authors refine the descriptions for more rigorous presentation. Once these adjustments have been made, I would recommend this paper for acceptance. Thank you for your efforts in addressing these concerns.
Author Response
Response to Reviewer
Dear Professor,
Thank you, once again, for your careful revision of our manuscript. The suggestions made in the previous revision were carefully accommodated in Section 4. We have no doubt that the quality of the paper was enhanced.
Thank you.
Yours sincerely,